# Ayu: a machine intelligence tool for identification of extracellular proteins in the marine secretome

Asier Zaragoza-Solas[1] ✉ & Federico Baltar [1,2] ✉

Microbes are the engines driving the elemental cycles. In order to interact with their environment and the community, microbes secrete proteins into the environment (known collectively as the secretome), where they remain active for prolonged periods of time. Despite the environmental relevance of microbes, our knowledge of the marine secretome remains limited due to a lack of effective in silico methods for the study of secreted proteins. An alternative approach to characterise the secretome is to combine modern machine learning tools with the evolutionary adaptation changes of the proteome to the marine environment. In this study, we identify and describe adaptations of marine extracellular proteins, which vary between phyla, resulting in differences in ATP costs, amino acid composition and nitrogen and sulphur content. We develop 'Ayu', a machine prediction tool that does not employ homology-based predictors and achieves better and quicker performance than current state-of-the-art software. When applied to oceanic samples (Tara Oceans dataset), our method was able to recover more than double the proteins compared to the most widely used method to identify secreted proteins. The application of this tool to open ocean samples allows better characterisation of the composition of the marine secretome.

The marine environment is the stage for critical geochemical processes key for maintaining planetary habitability, such as the production of atmospheric oxygen[1], the remineralization of organic carbon[2] and the cycling of nitrogen, phosphorus and sulphur[3–5]. Microbes have a central role in this process, as their genomes code for the enzymes that catalyse these chemical reactions. It is not surprising, then, that a large research effort has been poured into understanding the capabilities of marine microbe communities and their metabolic and genomic capabilities, resulting in global sampling expeditions like GEOTRACES[6], *Tara Oceans*[7] and Malaspina[8].

However, in our quest for understanding *what* microbes are capable of, the question of *where* seems to be key. Studies in the laboratory with pathogenic bacteria indicate that up to 30% of the bacterial genome encodes proteins that are released to the

extracellular milieu[9,10]. This subset of proteins (the secretome) is how bacteria interact with their environment, and as such are involved in a myriad of processes such as provision of nutrients through recognition, degradation and uptake of large extracellular molecules; communication and competition between bacterial cells; bacterial adhesion and biofilm production[11,12]. Exoproteome composition is also a key mechanism for bacterial adaptation. Studies of the exoproteome of *Ruegeria* and *Synechococcus* strains revealed that the exoproteome composition of each strain reflected their adaptations to their ecological niche and their growth conditions[13,14]. These discoveries have been corroborated in metaproteomics studies on both epipelagic and bathypelagic seawater samples[15,16]. Measurements of the extracellular enzymatic activity in the ocean indicate these reactions are largely catalysed by dissolved (cell-free) enzymes, with this ratio of cell-free to

[1]Fungal and Biogeochemical Oceanography Group, Department of Functional and Evolutionary Ecology, University of Vienna, Djerassi-platz 1, 1030 Vienna, Austria. [2]Shanghai Engineering Research Center of Hadal Science and Technology, College of Marine Sciences, Shanghai Ocean University, Shanghai, China. ✉e-mail: Asier.zaragoza.solas@univie.ac.at; fbaltar@shou.edu.cn

cell-attached enzymatic activity increasing with depth[17,18]. Furthermore, these experiments have also shown that the effects of the exoproteome can extend beyond the cell that secreted them, as exoproteins present a half-life of up to 20 days away from the cell[19], suggesting that in order to understand the nutrient utilisation capabilities of a marine community, the history of the water mass might be more important than the present community composition[18].

Despite the relevance of the secretome, its study is limited by the lack of appropriate methodology. Most marine prokaryotes cannot be cultured in lab conditions, and although shotgun metaproteomic approaches have been used to study the exoproteome, their throughput is low compared to proteomic assays based on bacterial cultures[15] and most of the material recovered belongs to virions[15,20]. Even proteomics assays in controlled environments present difficulties, as without careful methodology and quantification, it is difficult to determine which proteins found in the exoproteome are secreted or merely a product of cell lysis[21]. A reasonable approach would be to exploit the vast amounts of metagenome and metatranscriptome datasets available, but we are faced then with the challenge of predicting subcellular localization from the amino acid sequence.

Although the popularity of artificial intelligence and machine learning has led to the development of many tools for this purpose, a recent review by Hui et al.[22] compared several state-of-the-art available tools for Gram-negative bacteria and argued that "More enthusiasms have been put in new algorithms rather than the biological side" [sic]. Many of these tools are too narrow in their scope, focusing only on one secretion system or one bacterial strain, which limits their use in metagenomic data. Furthermore, many of these tools are only available to the scientific community as web servers, which are not suited to the high volumes of data required for omics datasets. Even pSORTb 3.0[23], the gold standard for subcellular localization predictions, relies on homology searches against its curated profile and protein databases to obtain its predictions, which severely limits the throughput of proteins it is able to process. It is no wonder, then, that most publications studying the secretome in omics datasets almost exclusively use SignalP[24] (a mature and robust software to detect Sec/Tat signals) to identify secreted proteins[16,25], even though translocation through the general secretion pathway does not guarantee secretion to the extracellular milieu, and many of the proteins with signal peptides stay in the periplasm or attached to the outer membrane[26].

Yet, the peculiarities of the marine environment present an opportunity to improve protein localization predictions. It is known that the amino acid composition (AAC) of a protein is in part adapted to the physicochemical properties of its location[27]. Seawater exhibits many distinguishing features, chief of which is its average salt concentration of 3.5%[28]. Salt has a denaturing effect in non-adapted proteins, mainly attributed to the disturbance of the water layer surrounding the protein, lowering solubility and promoting protein aggregation[29,30]. Cytoplasmic proteins are protected from the effects of salts, as marine bacteria are salt-out strategists which maintain a relatively salt-free cytoplasm[31]. However, this is not the case for proteins which operate in the periplasm, which is not osmoregulated[31–33], or in the extracellular milieu. Consequently, both extracellular and periplasmic proteins must be adapted to this salt content. Previous studies support this point: an analysis of proteomes found a correlation between isoelectric point (pI) and salinity[34], a survey of the proteins of the halophile gammaproteobacteria *Chromohalobacter salexigens* revealed that only the periplasmic and secreted proteins presented adaptations to salt[35], and a study comparing close phylogenetic neighbours of freshwater-saltwater pairs found differences in the isoelectric points and AAC of their encoded proteins, with differences being more pronounced in secreted than in cytoplasmic proteins[36].

Hence, in this study we first characterise the specific adaptations of extracellular proteins to the marine environment. Then, with this information, we developed a machine learning tool ('Ayu'), designed to exploit the signal left by these adaptations to predict secreted proteins in large marine metagenomic datasets, comparing its performance to state-of-the-art tools for subcellular location prediction. And finally, we applied this tool to environmental samples for the Tara Oceans expedition to uncover how much and which proteins composed the actual marine secretome.

## Results

### Differences in AAC between subcellular localizations and habitat

A biplot of the weighted log-ratio analysis of grouped AACs for marine proteins can be found in Supplementary Fig. S1. Although this collection of logratios does not explain a majority of the variance (42.5% within the first two components), the plot shows a gradient of changes from cytoplasmic to extracellular proteins, with periplasmic proteins situated between them. The loadings from the plot can be used to determine which amino acids are driving these differences: i.e., extracellular proteins are relatively enriched in polar (Ser, Thr, Asn, Glu), negatively charged (Asp, Glu) and aromatic (Phe, Tyr, Trp) amino acids. In contrast, cytoplasmic proteins contain more positively charged (Arg, Lys, His) and small hydrophobic (Val, Ile, Leu, Met, Cys) amino acids.

However, these changes could just reflect the different physicochemical properties of the cytoplasm compared to the other subcellular locations, and not to any specific effect of the extracellular environment. Therefore, we compared the AAC of non-marine proteomes to our marine representatives. As our control group, we chose the ESKAPEE pathogens (*Enterococcus faecium*, *Staphylococcus aureus*, *Klebsiella pneumoniae*, *Acinetobacter baumanii*, *Pseudomonas aeruginosa*, *Enterobacter spp*. and *Escherichia coli*), as they are an extensively studied group of bacteria[37], and none of their members are commonly found in marine environments. Figure 1A shows radar plots comparing the AAC of our marine dataset and the proteome of the seven ESKAPEE pathogens. While both groups follow the systemic differences previously described for the marine proteomes, further differences between both groups of proteomes can be observed depending on the subcellular location (Supplementary Fig. S2). While cytoplasmic proteins from both groups present a similar composition, periplasmic and extracellular marine proteins show an increase in negatively charged and aromatic amino acids, and a decrease in some positively charged (Arg, Lys) and hydrophobic (Ala, Leu, Val) amino acids, compared to ESKAPEE proteins found in the same subcellular location.

To test if these differences were statistically significant, a Dirichlet regression analysis was performed, adjusting for subcellular localization and cell wall structure (that is, if the proteins were coded by a gram positive or gram negative bacterium). The complete results of the analysis can be found in Supplementary Text S1. The effect of habitat was modelled as an interaction between habitat and subcellular location, with the objective of assessing how each cellular compartment was affected separately. This interaction was found to be statistically significant by comparing the full model to a reduced model with the interaction removed (difference in deviance = 1794, p-value < 2.2e −16). Overall, the model indicates that while subcellular location and cell wall structure influence AAC, these differences are intensified for marine proteins relative to ESKAPEE proteins, especially for periplasmic and extracellular proteins. Amino acids E, G, I, K, P, T, V are especially affected, with their effects increasing at least 20% (p-value < 1e−5 for all comparisons). Additionally, the marine environment affected periplasmic and extracellular proteins differently, as the AAC of extracellular proteins was influenced more strongly towards polar, aromatic and charged amino acids. When considering both the changes in AAC explained by subcellular location and the influence of the marine environment, we find that in 12 amino acids (L, V, F, Y, S, T, N, Q, D, E, K, R) the distribution varies following their order of

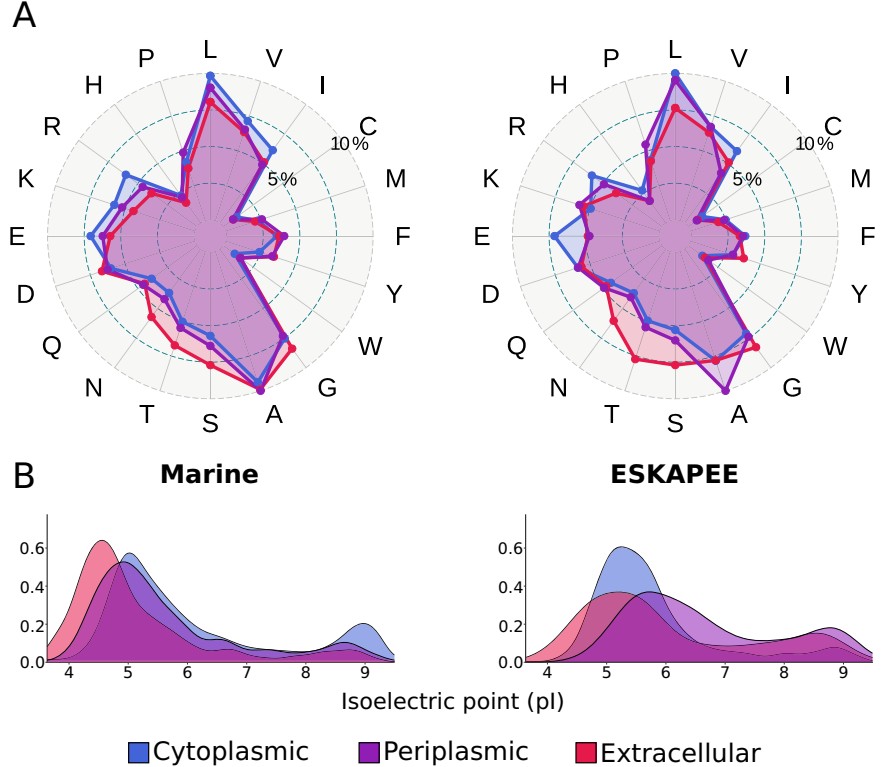

**Fig. 1 | Differences in amino acid composition and pI based on habitat. A** Radar plots of amino acid composition for the marine (left) and ESKAPEE (Right) protein datasets. **B** Density curves for the isoelectric point (pI) distribution for the marine (left) and ESKAPEE (Right) protein datasets. In both subplots, proteins are separated by subcellular location.

exposition to the environment (Extracellular > Periplasmic > Cytoplasm), compared to 7 in the non-marine ESKAPEE proteins (L, Y, G, S, T, N, R).

Overall, these results prove that the marine environment has a specific effect on the proteins exposed to it. However, the question of which effectors might force these constraints still stands. Salinity is the obvious culprit, as it both has a well-known effect in protein function and it is known that bacterial cytoplasms are relatively salt-free compared to the periplasm and the extracellular milieu[35,38]. In fact, the pattern of amino acid substitution observed in marine extracellular proteins (Decrease in positively charged and aliphatic amino acids, increase of negatively charged and small amino acids) is remarkably similar to that reported for salt adaptation in halophilic proteins[31]. The increase in aromatic residues is the exception to the rule, but there are studies reporting a link between hydrophobic interactions between aromatic residues and enzyme halotolerance[39], possibly by forming weak polar interactions with other residues[40,41]. Another possible explanation is that the increase in aromatic residues is a product of an adaptation to multiple stresses. In that respect, there are multiple reports where the addition of aromatic residues to a marine protein increased its tolerance to high and low temperatures[39,42,43].

### Differences in AAC between taxonomic groups

As the Dirichlet regression model indicated a significant taxonomy effect in AAC, we inspected the proteomes of a few taxa with enough proteins ($n > 30$) in each location to provide a fair estimate. Supplementary Fig. S3 shows a Log Ratio Analysis biplot that exposes this interaction between taxonomy and AAC, with differences between phyla caused by the proportion of positively charged (His, Arg, Lys) and small apolar amino acids (Ala, Gly). Although the amount of proteins available only allow for comparisons at the level of order, we can dig deeper by performing analyses on groups of interest.

Supplementary Fig. S4 shows the amino acid distribution in the orders Synechococcales, Flavobacteriales, Bacilliales and Micrococcales. Comparing the four sets of extracellular proteins, it can be observed that while all groups follow the trends described previously (i.e., increase in negatively charged and polar residues compared to positively charged and hydrophobic residues as we increase exposure to the extracellular milieu) each order follows its own amino acid distribution. For example, Synechococcales adapt to the extracellular medium by increasing the ratio of Glutamic acid at the expense of polar amino acids, while extracellular proteins in Flavobacteriales and Rhodospirillales (Supplementary Fig. S5) are enriched in polar amino acids.

To explain these differences in AAC across phyla, we might turn to the differences in lifestyle between the shown taxa. Previous studies have speculated that extracellular proteins produced by bacteria will be on average cheaper than their cytosolic counterparts, as these proteins cannot be recycled, and because cooperation is more likely to evolve when the costs of said cooperation are outweighed by the advantages provided by their kin in the community[44]. Here, we have found that the cost of extracellular proteins varies significantly between phyla. We propose that these differences originate from the different trophic strategies of the bacteria that produce them. It is not unreasonable that *Alteromonas*, a copiotrophic bacteria with chemotaxis and motility to detect locations with high nutrient availability[45], will get a better return for their investment in the production of extracellular enzymes than an oligotrophic organism such as *Prochlorococcus*, which can not guarantee that its extracellular proteins will provide a profit[46]. Extracellular enzymes are an important part of bacterial adaptation to their ecological niche, as their genetic properties reveal: extracellular protein repertoire is highly divergent compared to intracellular proteins[10,13], and they tend to be lost and gained at a high rate in bacterial genomes[47,48]. As different trophic strategies

will be under different selection pressures, it would stand to reason that extracellular proteins from oligotrophic organisms are selected for cost.

## Differences in protein properties derived from AAC

As AAC is the basis for all characteristics of a protein, it would be reasonable to expect that the differences between subcellular locations be reflected in other protein properties. The classical example of this would be the isoelectric point (pI), which is a combination of the dissociation constant (pKa) values of the constituent amino acids[49]. In fact, it is already well known that bacterial proteomes present a bimodal distribution, with both peaks corresponding to cytosolic and integral membrane proteins[49]. In the marine dataset, we found an evident shift towards the acidic end of the scale as the location gets closer to the extracellular milieu (Fig. 1B). These results are consistent with those found in proteomes of closely-related bacteria lineages that inhabit either freshwater or marine waters[36].

Another property that might help distinguish between secreted and non-secreted proteins is their cost, as extracellular proteins tend to be composed of amino acids that are less expensive to produce[47,50]. We calculated the average ATP cost, average nitrogen and sulphur content for the marine dataset and found mixed results. On the one hand, there is a small (effect size 0.0121) but significant (p-value < 0.00483, kruskal–wallis test) reduction in ATP cost for both extracellular and periplasmic relative to cytoplasmic proteins. However, the magnitude of the effect is highly dependent on phylum, confirming previous reports[51]. In the marine dataset, only Synechococcales, Flavobacteriales and Rhodospirillales show a large reduction in ATP cost (Supplementary Fig. S6, effect size >0.2 for all three groups). This provides further support to the observation mentioned above -that differences in AAC between taxa are related to trophic strategy. With regards to nitrogen and sulphur content, there is a clear decrease of these elements in extracellular proteins compared to their intracellular counterparts (p-value < 8.29E−193, effect size = 0.1 for sulphur and 0.178 for nitrogen) (Supplementary Fig. S7). However, there is a decrease in average carbon content in extracellular proteins compared to the other groups, probably as a side effect of amphiphilic amino acids being replaced by the smaller amino acids alanine and glycine (Supplementary Fig. S7).

It has been reported that in the gut microbiota, extracellular proteins are longer than their intracellular counterparts[51]. We confirm that this observation stands in our marine dataset (Supplementary Fig. S7). Interestingly, this increase in length is not accompanied by an increase in molecular weight, as the amino acids more prevalent in extracellular proteins tend to have smaller side chains. A possible explanation for this phenomenon is that larger proteins present less diffusion length, therefore increasing the time that the protein stays close to the bacteria. Garcia-Garcera et al.[50] discovered that bacteria inhabiting poorly structured habitats tend to produce protein with lower diffusion lengths, as there is no community to share the burden of producing extracellular effectors. In this paper, diffusion length was calculated with the Einstein-Stokes equation, in which diffusion length is a function of temperature and molecular weight. We found no significant differences in molecular weight, but perhaps more sophisticated methods to calculate protein diffusion rates, such as HYDROPRO[52] or HullRad[53] might uncover differences in this property between extracellular and cytoplasmic proteins Another, more parsimonious explanation is that secreted proteins are larger since they need to incorporate sorting signals in their sequence.

## Sequence order effect

A major drawback of composition-based protein descriptors is that they ignore the distribution of amino acids in the protein sequence. As it has been demonstrated that including this information in a prediction model can significantly increase its performance for cellular

location prediction tasks[54], it is of interest to assess their possible contributions to subcellular localization prediction. The most basic descriptor of sequence order would be di/tripeptide composition, as it incorporates neighbourhood information. The top 40 dipeptides from our marine dataset that vary the most between subcellular localizations are shown in Supplementary Fig. S8. As a general rule, most of these dipeptides are combinations of the single amino acids with the most variation, but it includes some unusual pairings that can help distinguish between cellular locations. An example of this are the dipeptide pairs NP and PN, which are most abundant in extracellular and periplasmic proteins.

In order to assess sequence order over larger distances, a commonly used approach is to calculate autocorrelation in the sequence (that is, how similar the sequence is to itself by given intervals), after translating each amino acid to a numerical value provided by a propensity scale, such as those provided by AAIndex[55]. Although these descriptors are less intuitively interpretable and capture less information than 3D-based protein analyses, they still capture information not detected by amino acid composition. For example, autocorrelation plots have been found to reflect the tertiary structure of a protein[56] and have been used to predict secondary structure composition[57–59]. Furthermore, they are much faster to calculate and only require the amino acid sequence. As the effect of the chosen propensity scale is marginal at best[60], we decided to only use two types of autocorrelation descriptors: partial Quasi Sequence Order (pQSO), an adaptation of QSO[61] which uses the physicochemical distances between pairs of amino acids calculated by Schneider and Wrede[62], and Pseudo amino acid composition (pPAAC), an adaptation of Chou's PAAC metric[63] which combines hydrophobicity, hydrophilicity and residue mass propensity scales. A non-parametric MANOVA analysis of the ILR-transformed features reveals that although each variable has a small effect size (relative effects mean = 0.462, std = 0.025 for pQSO, relative effects mean = 0.503, std = 0.026 for pPAAC) and a lot of variability, the combination of at least 20 autocorrelation measures is enough to reliably distinguish between subcellular locations (p-value < 1.5E−20 for both QSO and PseAAC). As the correlation between these two sets of features was small (<0.3, Pearson correlation), we included both sets of predictors in the final model.

## Machine learning model design and validation

With a validated set of protein descriptors, we tested if this information could be used to improve on current subcellular location prediction methods. xgBoost was our algorithm of choice since it presents many advantages: can be used with non-parametric data, supports multi classification, is reasonably resistant to overfitting if the right hyperparameters are used and it has been used extensively for protein classification problems[64,65]. As our analysis had revealed a gradient of adaptations with the order extracellular > periplasmic > cytoplasmic, we suspected predictions would improve by framing the problem as an ordinal classification. Therefore, two classification strategies were implemented: a multiclass classifier, treating each subcellular location as an independent class, and an ordinal classifier, which is aware of the intrinsic ordering between the classes.

The results of the five-fold cross-validated testing of both Ayu implementations (ordinal and multiclass), pSORTb 3.0[23] and BUSCA[66] are collected on Table 1. In general, when comparing MCC and Kappa scores, all Ayu implementations (MCC > 0.89, Kappa > 0.89 for all implementations) significantly overperform compared to pSORTb3 (MCC = 0.64, Kappa = 0.64) and BUSCA (MCC = 0.43, Kappa = 0.42). Examining the precision-recall metrics for each group (Fig. 2A), it is clear that while pSORTb3 obtains a reasonable precision in all three subcellular localizations (0.97 for cytoplasmic predictions, 0.9 for periplasmic, 0.93 for extracellular), its recall score is fairly low for periplasmic (0.58) and extracellular (0.46) proteins; meaning pSORTb3 mis-classifies more than half of the non-cytoplasmic proteins

**Table 1 | Classification metrics for Ayu, pSORTb3 and BUSCA**

| | Precision (Cyto) | Recall (Cyto) | Precision (Peri) | Recall (Peri) | Precision (Extr) | Recall (Extr) | MCC | Kappa |
|---|---|---|---|---|---|---|---|---|
| psortb3 | 0.96 | 0.96 | 0.88 | 0.58 | 0.91 | 0.46 | 0.64 | 0.64 |
| BUSCA | 0.93 | 0.9 | – | – | 0.44 | 0.61 | 0.43 | 0.42 |
| Ayu (Multiclass) | 0.97 | 0.97 | 0.88 | 0.87 | 0.86 | 0.83 | 0.89 | 0.9 |
| Ayu (Multiclass, SMOTE) | 0.98 | 0.99 | 0.89 | 0.85 | 0.89 | 0.81 | 0.91 | 0.9 |
| Ayu (Ordinal) | 0.97 | 0.99 | 0.91 | 0.82 | 0.93 | 0.7 | 0.89 | 0.89 |

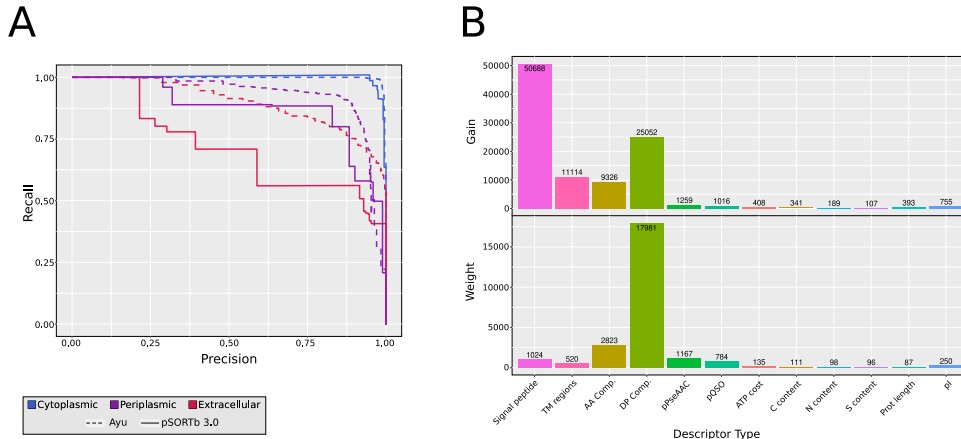

**Fig. 2 | Performance of Ayu compared to other classifiers. A** Precision-recall curve for Ayu multiclass (dashed line) and pSORTb3 (solid line), separated by cellular location. **B** Feature importance values for Ayu multiclassifier and all protein descriptors: gain (top) and weight (bottom).

as false negatives. BUSCA only achieves good scores for cytoplasmic proteins (precision = 0.93, recall = 0.9), presenting low precision and recall scores for extracellular proteins (precision = 0.44, recall = 0.61). The reason for a worse than random precision score is probably due to the fact that BUSCA only uses signal peptide information to predict the extracellular localization, therefore including periplasmic proteins that also contain a signal peptide.

Both versions of Ayu (multiclass and ordinal) present an improvement over the other classifiers, although with differences in recall and precision. The ordinal version presents better precision scores than both multiclass versions (0.97/0.92/0.93 Ordinal, 0.97/0.91/0.89 Multiclass for Cytoplasmic/Periplasmic/Extracellular), but the multiclass version achieves better recall (0.99/0.82/0.7 Ordinal, 0.99/0.85/0.81 Multiclass for Cytoplasmic/Periplasmic/Extracellular).

Likewise, the application of the SMOTE algorithm to ameliorate the imbalance between protein classes results in a small improvement in the multiclass implementation of Ayu (MCC score improvement of 0.02). As the use of SMOTE only affects training time does not impact prediction time, the SMOTE version of the multiclass implementation was kept for the final version of Ayu, as we consider the tradeoff between recall and precision to be better in the multiclass version.

As xgBoost belongs to the algorithm family of boosted trees, we are able to obtain feature importance scores, which contain information about which feature descriptors are more useful to discriminate between classes. Figure 2B shows the permuted feature importance results for the Ayu SMOTE multiclass version. "Gain" represents the contribution of each feature to the classification, while "Weight" indicates how many times the feature appears in a tree across the ensemble of trees. Therefore, the graph shows that while Signal peptide information is by far the most important feature to discriminate between the three cellular locations, Dipeptide composition is the feature that appears most in the trees, probably due to the fact that it includes more features.

However, this does not mean that all features are equally important for the classification to all locations. An example of this can be found on Supplementary Fig. S9, which plots the permuted feature

importance gains for the individual binary classifiers in Ayu Ordinal. For the cytoplasmic vs non-cytoplasmic predictor, the most important protein descriptors are the presence of a signal peptide, the ratio of transmembrane to non-transmembrane regions in the protein and the dipeptide composition. On the other hand, the classifier tasked with discriminating between periplasmic and extracellular proteins finds that amino acid and dipeptide composition are more important than the presence of a signal peptide, corroborating the magnitude of the differences between proteins in different subcellular locations.

### Application to real world (marine) dataset: Tara Oceans

In order to test the performance of Ayu in a real world metagenomic dataset, we applied our prediction tool on 6 Tara Oceans metagenomic and metatranscriptomic datasets, comprising three sampling sites of the prokaryotic fraction at surface and mesopelagic depths of different ocean basins (Supplementary Data S1). Out of the 46,775,154 total proteins found in the combined dataset, 73% of the sequences belong to bacterial genes, 8% to viral genes and 3% to archaeal genes, with the rest having no taxonomic classification. The results of the subcellular location classification for bacterial proteins with Ayu are summarised in Fig. 3A. Around 15.7% of the proteins were classified as transmembrane proteins by manual classification (see Methods). Out of the remaining proteins, 65.2% are classified as cytoplasmic, while 12.5% of proteins are classified as non-cytoplasmic (5.5% extracellular, 7.0% periplasmic). 5.6% of the proteins are not classified into any category. The proportion of non-integral membrane proteins secreted into the periplasm or to the extracellular milieu (12.5%) reported here falls in the range of the size of the secreted proteome reported in previous studies (10–30%)[67,68]. Interestingly, even though Ayu was trained only with bacterial proteins, the predictions for viral proteins are accurate, with DNA replication and auxiliary metabolic genes classified as cytoplasmic and virion structural proteins classified as periplasmic or extracellular. As virion proteins are in contact with the same extracellular milieu as bacterial extracellular proteins, this is further proof that our tool is using adaptations to the extracellular conditions to classify proteins.

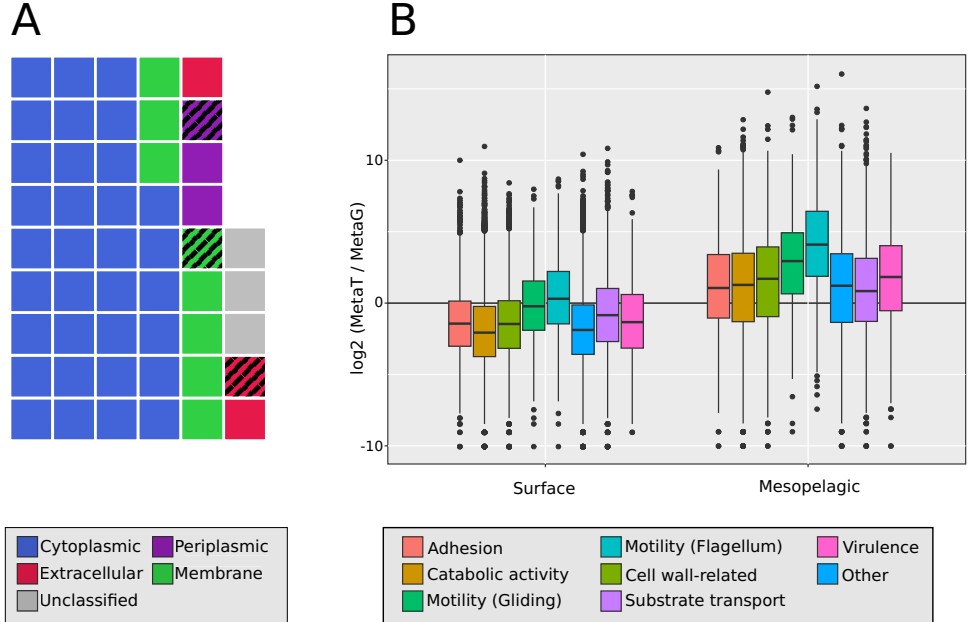

**Fig. 3 | Extracellular protein function in Tara Oceans dataset. A** Waffle chart comprising the subcellular localization distribution for bacterial Tara oceans proteins. Each square represents 2% of the total (for precise ratios, see text). A striped square indicates that fraction codes for a signal peptide. **B** log2 ratios of RPKG in metagenome vs RPKG in transcriptome for the Tara Oceans bacterial extracellular fraction, separated by function (*n* = 8674). In the box plots, the black bar indicates the median, the range of each box extends from the first to the third quartile, and whiskers extend to 1.5-fold interquartile range. Protein recruitment values are pooled from three different samplings performed by the Tara Oceans expedition, taken at two different depths. Full information of the samples used can be found in Supplementary Data S1.

As Ayu uses signal peptide information as one of its features, we are able to ascertain how many proteins predicted to each cellular location include a signal peptide that is detected by SignalP[24]. For bacterial proteins, only 79% of periplasmic and 54.7% of extracellular proteins contain a signal peptide (Fig. 3A), reflecting the importance of GSP-independent secretion systems in the marine secretome, and the relevance of Ayu. To test the prevalence of these cases, we clustered proteins at 70% identity with 95% coverage (see Methods), presuming that proteins that cluster together should contain the same secretion signals. Our results show that out of 53,902 proteins grouped in clusters with at least 1 protein with a signal peptide, only 43,361 (80%) coded for one. These results suggest that Ayu is able to complement signal peptide prediction to recover more intra-cluster extracellular protein diversity.

The aforementioned clustering process also produced several protein clusters of predicted extracellular proteins without a signal peptide. These clusters sum up to 39,871 proteins, representing almost half of the total extracellular proteins detected in this dataset. Although only 53% of the proteins detected in this way can be annotated, it is still possible to find proteins that further prove the validity of the prediction method. Interestingly, while the extracellular proteins with a signal peptide are dominated by catabolic enzymes (peptidases, glycosyl hydrolases, sulfatases, phosphatases) and outer membrane-associated proteins (assembly factor BamB/BamE, TonB-dependent receptors), the bulk of the signal peptide-free protein fraction is composed of flagellins (FliC/FljB/LafA), flagellar basal body rod proteins (FlgC/FlaE/FlgF) and flagellar hook-associated proteins. These proteins are secreted via the T3SS pathway, which translocates proteins directly from the cytoplasm[69]. Proteins with secretion signals for other pathways can also be found, including proteins with RTX repeats (involved in secretion via T1SS)[70] and domains related to secretion pathways II, III, V, VI and VII. Surprisingly, some proteins contain sorting domains related to T9SS, a widespread secretion pathway in Bacteroidetes that is dependent on the GSP for translocation into the periplasm[71]. The absence of signal peptide in these proteins suggests that either these proteins reach the periplasm via another method or the signal peptide used is not detected by SignalP6. A group of extracellular proteins of particular interest are those related with interaction with other members of the microbial community. Examples of this group are an homolog of the protein Reb, involved in interactions between bacteria and eukaryotes[72]; and several proteins encoding the Nif11 domain, found in the leader peptide of various bacterial microcins[73]. These results provide further evidence for the predictions of Ayu to not be mere false positives.

Finally, we studied and compared the metagenomic to the metatranscriptomic dataset from the same Tara Oceans samples to test for differential patterns based on gene content or expression. Overall, there was a relatively high expression of genes identified as coding for secreted proteins, confirming the relevance of the secretome in the environment. We found that the ratio of metatranscriptome RPKG (Reads Per Kilobase of sequence per Gigabase of dataset) to metagenome RPKG is an order of magnitude larger in mesopelagic samples compared to surface samples (log2 fold metaT/metaG in surface = −0.93, log fold metaT/metaG in mesopelagic = 0.84) (Supplementary Fig. S10), which is consistent with previous studies reporting protein activity and transcription to increase with depth[16,17]. As a general rule, metagenome samples have a wider spread of genes in the gene pool (8674 genes with >1 RPKG in surface samples, compared to 7292 in mesopelagic), but expression of the genes present is higher in mesopelagic samples. We have found differences between taxonomic clades (Supplementary Fig. S11). For example, genes assigned to the orders Alteromonadales, Vibrionales and Flavobacteriales show the previously reported increase in mesopelagic transcription (effect size >0.4 for all three orders), while those genes assigned to orders Pelagibacterales, Synechococcales and Bacillales show a lower increase (effect size = 0.17 for Synechococcales). These results are consistent with previous studies reporting Gammaproteobacteria and Flavobacteriales as the main producers of extracellular proteins in the mesopelagic[16]. Finally, we found no clear distinction in gene functions between surface and mesopelagic samples (Fig. 3B), which is also consistent with previous studies reporting high functional redundancy of microbial functions with depth[16,19].

## Discussion

In this work, we have shown that the marine environment has a significant effect on the proteins that must operate in that environment, and that the imposed constraints in amino acid composition allow for discrimination of bacterial proteins based on their subcellular location. These differences are the basis for the tool presented in this paper, which surpasses the performance of current methods for proteins sourced from the ocean. Ayu also presents a series of advantages aside from its performance. It only uses signal peptide and transmembrane regions as external protein descriptors, relying on sequence-based descriptors for the rest, meaning it will remain useful for a longer time than homology- and PSSM-based methods, which must be constantly updated with new discoveries in order to stay accurate.

However, this reliance on amino acid adaptations to the environment means that great care should be put in its use in order to avoid spurious classifications: Ayu was not trained with membrane proteins, so predictions will likely be spurious for transmembrane proteins. Cell wall-attached proteins, such as those with the LPXTG motif[74], will be predicted as extracellular or periplasmic depending on their location. Likewise, the program might have issues with proteins that are found in minor subcellular locations, such as the thylakoids found in cyanobacteria, as the particular physicochemical conditions of said organelles is different from the cytoplasm[75] might affect amino acid composition. Additionally, we would recommend only using Ayu for prokaryotic and bacteriophage genomes. Eukaryotes possess different subcellular locations than prokaryotes, and signal peptides are not only employed for outer membrane translocation, but also for protein trafficking between organelles[76]. Predictions in archaea should work, as signal peptides also only control translocation through the cytoplasmic membrane[77,78]. However, the salt-in strategy is more widespread in archaea than in prokaryotes[79], and very few marine archaea have been isolated in order to test their salt strategy[80].

Collectively, this study combined the study of genomic, transcriptomic, proteomic and ecological properties of marine microbes with recent advances in artificial intelligence to push further the limits of our knowledge of the secretome and thereby of marine biology and biogeochemistry. With this approach we have discovered that we were missing at least half of the story (i.e., doubled the size of the secretome), which becomes particularly relevant in the light of climate change, where the activity of microbes is expected to play a key role particularly through their secretome. We expect that the use of this tool will shed light on this key but poorly understood area of marine biology and biogeochemistry, particularly in areas beyond catabolic activity, which have been relatively more studied[16]. For instance, the presence of microcins and other proteins involved in the interaction between members of the microbial community is of particular interest, as it represents an untapped pool for the discovery of novel antibiotic factors. Finally, the approach used in this study, of combining biological/ecological adaptations of proteins to artificial intelligence tools, can be applied to other environments with different environmental conditions/adaptations, further pushing the frontiers of knowledge of the ecosystem services provided by microbes on Earth, and how they might be affected by environmental changes.

## Methods

### Data collection, annotation and curation

A dataset was compiled to both study the adaptations of bacterial proteins to the marine environment and train a machine learning tool. An overall flowchart of the data collection process can be found in Supplementary Fig. S12. First, an extensive bibliographic search was conducted to select a collection of bacteria that met all of the following characteristics: a) the bacteria had been isolated from the marine environment, or its genome sequence assembled from a marine sample; b) the bacteria was a mild halophile, as defined in refs. 38,81, or had been only described in marine environment. This process resulted in a collection of 105 Gram-positive and 929 Gram-negative bacteria (Supplementary Data S2). Protein sequences were then downloaded from UniProtKB[82] or the NCBI Genomes database[83].

The gold standard for protein datasets is the use of proteins with experimentally proven locations[23]. Unfortunately, out of the 17,047 proteins coded by our collection of marine bacteria, only 38 of them have their location experimentally proven, and as we are interested in studying the adaptation to the marine environment, we classified the proteins following these criteria. First, proteins with a reviewed cellular location annotation in UniProtKB were divided into three groups (cytoplasmic, periplasmic and secreted). Moonlighting proteins (sequences reported to be present in more than one subcellular location) or proteins with ambiguous annotation were removed from the dataset.

According to previous reports[84,85], proteins that share function and have at least ≥25% global sequence identity tend to share subcellular location. Unreviewed marine proteins were aligned to a subset of prokaryotic proteins with manually curated subcellular locations downloaded from UniProtKB ("reference dataset", 1,131,685 proteins) using ggsearch[86]. For each marine protein with a match to a reference sequence that passed the 25% identity threshold, domains and features of both sequences were detected using InterProScan[87], and the cellular location from the reference sequence was propagated to the marine one only if the content and synteny of domains and features was the same for both proteins. Finally, proteins were clustered to 30% sequence identity and coverage alignment of at least 80% using CD-HIT[88]. From each cluster, the protein with the best match to a reference sequence was selected to be part of the training dataset. The resulting dataset contains 17,047 proteins (1934 Gram-positive, 15,113 Gram-negative), divided into 14,410 cytoplasmic, 1873 periplasmic, 764 extracellular proteins.

Differences in signal peptide content is a defining characteristic of the different locations, as by definition cytoplasmic proteins will never code for one, while proteins the other two categories might. Therefore, we performed a circularity test for the dataset as defined by Riezler and Hagmann[89]. Briefly, two Generalised Additive Models (GAMs) were fit to the dataset, one including signal peptide information as an explanatory variable (GAM_wSP) and one without (GAM_woSP). The tests show that the dataset does not meet any of the two criterions established: although the scaled deviance is remarkably higher in GAM_wSP than in GAM_woSP ($D^2 = 0.524$ vs $D^2 = 0.352$), it is not close to the value of near 1 that we would expect if the signal peptide information was able to predict the entire dataset. Supplementary Fig. S13 shows that features in both GAMs are still contributing to the classification of the model, as evidenced by the fact that they present a non-zero feature shape. Therefore, the second criterion (nullification of the contributions of other features) can be also ruled out.

From this compiled protein dataset, 70% of the proteins (stratified by subcellular location) were reserved for data exploration, feature extraction and model training, while the remaining 30% were only used for further evaluation of the model (15% for validation during training, 15% for testing).

In order to test if these amino acid adaptations were only found in the marine environment, the ESKAPEE group (*Enterococcus faecium*, *Staphylococcus aureus*, *Klebsiella pneumoniae*, *Acinetobacter baumanii*, *Pseudomonas aeruginosa*, *Enterobacter spp.* and *Escherichia coli*) were used as a control group. This group was chosen as they are an extensively studied group of bacteria, due to their clinical importance[37], and none of its members are typically found in marine open waters. Proteins from these organisms were downloaded from PSORTdb 4.0[90], including both experimentally-proven and predicted subcellular locations.

### Statistics and compositional analysis

Standard statistical analyses (Kruskal–Wallis test, posthoc Dunn's test, eta squared effect size calculations) were performed using the R

package statix[91]. Non-parametric MANOVA was performed using the R package npmv[92]. As many of the feature datasets employed in this paper are compositional, that is, data that carry only relative to a total, special care has been taken to use compositionally-appropriate methods when necessary. When interpretability of the results was not a concern, the scikit-bio package[93] was used to treat compositions first with multiplicative replacement to remove all zero values, then an isometric logratio (ILR) transformation was applied to the composition, in order to transform the data into linearly independent, sub compositionally coherent features that can be analysed with standard statistical methods[94]. The exception to this is the compositional data obtained from TMBed[95]. As we were only interested in the ratio between transmembrane and extramembrane components, the composition was modelled as an amalgam logratio with the following formula (1):

$$SLR = \log\left(\frac{pH + pB}{pS + po + pi}\right) \quad (1)$$

In which $pH$, $pB$, $pS$, $po$, $pi$ represent the proportions for transmembrane alpha helices, beta barrels, signal peptide, cytoplasmic and external regions respectively. As ILR-transformed data is significantly more difficult to interpret, we used non-transformative methods for data exploration. The weighted log ratio variance plot was calculated using the easyCODA package[96], while regression analysis was performed using the DirichletReg R package[97]. Full regression results can be found in Supplementary Text S1.

For the circularity check, the methodology presented in ref. 89 was applied. Generalized Additive Models (GAMs), feature shape plots and scaled deviance calculation were performed using the mgcv R package[98].

### Protein feature extraction

A list of the protein descriptors used in this paper can be found in Supplementary Data S3. Each protein descriptor was chosen by their capacity to discriminate between proteins located in different cellular compartments (see Results). Each protein is represented by a 466 length feature vector. All features were extracted using in-house scripts, following the methodology stated in their respective papers. Many of the protein features that account for sequence order include AAC in their feature list, introducing redundancy in the feature set. With the objective of including multiple sequence order features and removing redundant information from the feature dataset, slightly modified formulas for quasi sequence order (QSO)[61] and pseudo amino acid composition (PseAAC)[63] were employed. Descriptions of these modified formulas can be found in Supplementary Text S2. Transmembrane regions, signal peptide information and isoelectric points for each protein were also calculated and included as features, using the programs TMBed[95], SignalP6.0[24] and IPC2.0[99] respectively.

### Model training, optimization and validation

The features extracted in previous steps were used to train xgBoost models using the python package xgBoost[100]. Both a multiclass and an ordinal classifier were trained, with the latter being implemented following the scheme described in ref. 101. Briefly, N-1 binary classifiers are trained to predict N categories, and the probability for prediction of the middle categories is defined by the ensemble of the N-1 binary classifiers. For this specific application, two classifiers were trained, one to discriminate between cytoplasmic and periplasmic+extracellular proteins and another to distinguish between periplasmic and extracellular proteins (Supplementary Fig. S14).

Hyperparameters for each model were optimised with a five-fold cross validation grid search, as implemented in scikit-learn[102]. Early stopping of the fitting step was implemented to reduce the risk of overfitting. The train-test partition and the splits for cross-validation were performed using graphpart[103], an homology-aware partitioner, using the recommended 30% identity threshold. As stated previously, the uniprot training dataset is severely imbalanced for two of the three classes (1:1:10 ratio extracellular:periplasmic:cytoplasmic). As imbalanced ratios between classes might cause difficulties during the fitting process, an oversampling SMOTE algorithm, as implemented in the python package imbalancedlearn[104], was applied to the training split before fitting the multi class classifier. Feature importance analysis was also performed for all models to assess the performance of each feature group. Finally, fitted models were evaluated against the validation partition, using the metrics precision (Pr), recall (Rc), precision-recall curves, Matthews correlation coefficient (MCC), using Gorodkin's k-category definition[105] and Cohen's Kappa (Kappa).

### Comparison against other cellular localization classifiers

The performance of our tool was compared against other subcellular localization predictors (pSORTb 3.0 and BUSCA). The marine testing dataset was predicted using pSORTb 3.0[23] and BUSCA[66], as they are the two non-ensemble predictors that have the best reported performance to date[106] and are readily available. As BUSCA is only available as a web server with a limited throughput, an in-house script was used to run predictions. Performance metrics mentioned in the previous section were calculated for both methods per class using scikit-learn[102]. Precision-recall curves were only calculated for pSORTb, as BUSCA only provides probability scores for the predicted category.

### Sub localization prediction in Tara Oceans samples

To test the performance of the machine learning model in a real marine dataset, the proteins contained in Ocean Microbial Reference Catalogue v2 (OM-RGC.v2)[107] were downloaded and classified into different cellular locations using the machine learning models built in previous steps (See "Model training, optimization and validation"). The dataset was functionally annotated against the CDD database[108] profile database using hmmscan[109]. A profile hit was kept if the e-value < 1e−5 and the alignment covered at least 75% of the profile. Proteins from OM-RGC.v2 were clustered into protein clusters to 70% identity and 95% coverage using mmseqs2[110]. Membrane proteins were identified by calculating transmembrane regions for all proteins with TMBed[95], classifying a protein as transmembrane if it contained at least 5% of its protein length in a beta barrel or a transmembrane helix. Proteins with only one transmembrane helix on the N-Terminal were included as part of the membrane location only if they contained a domain with a GO term with located the protein to a membrane[111], as signal peptides can be misclassified as transmembrane helices[112]. Finally, recruitment values in metagenome and metatranscriptome samples provided in the OM-RGC.v2 dataset were used to compare gene abundance and gene transcription rates between protein subcellular locations and depth.

### Reporting summary

Further information on research design is available in the Nature Portfolio Reporting Summary linked to this article.

## Data availability

The UniprotKB IDs and NCBI GIs of the proteins analysed in this study and used to train and validate the model can be found in the Source Data files for this paper, Table S1. The IDs for Tara Oceans metagenomes and metatranscriptomes used in this study can be found in Supplementary Data S1. Source data are provided with this paper.

## Code availability

The code for the machine learning prediction tool described on this paper (named 'Ayu' in reference to the amphidromous fish), as well as a list of the proteins used for training and validating it, can be found with https://doi.org/10.5281/zenodo.14865847.

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

## Acknowledgements

This research was funded in whole or in part by the Austrian Science Fund (FWF) P35248. For open access purposes, the author has applied a CC BY public copyright license to any author accepted manuscript version arising from this submission. The computational results of this work have been achieved using the Life Science Compute Cluster (LiSC) of the University of Vienna.

## Author contributions

A.Z.-S and F.B. conceived the study. A.Z.-S. ran the experiments and analysed the data. All authors contributed to writing the manuscript.

## Competing interests

The authors declare no competing interests.
