## [Transparent Peer Review file · Nature Communications]

A novel artificial intelligence tool for identification of extracellular proteins doubles current estimate of the marine secretome

Corresponding Author: Professor Federico Baltar

Version 0:

Reviewer comments:

Reviewer #1

(Remarks to the Author)

General comment:

This manuscript describes a novel tool for predicting secreted proteins in marine bacteria. It is a very welcome addition to the prokaryotic protein sorting prediction toolbox, where the reference method, PSORTb 3.0, is more than a decade old. However, I have serious doubts about the separation between training and test data and hence about the reported predictive performance.

It is stated that 30% of the dataset were only used for evaluation of the model, and that "proteins were grouped into clusters with 50% sequence identity and at a coverage alignment of at least 80%, using the MMseqs2 software suite".

There are two problems with this. First, 50% identity and 80% coverage constitutes a rather high threshold; if this is compared to other recent subcellular location prediction methods (mainly in eukaryotes), 30% or 40% identity would be more appropriate. Second, MMseqs2 simply cannot be used in this way. Only the representative sequence of each cluster (as defined by MMseqs2) have a maximum identity of 50% to other representative sequences; this restriction does not apply to other members of the clusters. In other words, there may very well be sequences in the test set that are considerably more than 50% identical to sequences in the training set.

A proper separation between training and test can be achieved by either (i) reducing the dataset by only keeping the representative sequence of each MMseqs2 cluster, or (ii) using a homology partitioning scheme such as GraphPart (<https://doi.org/10.1093/nargab/lqad088>).

Specific comments, major:

- I.88-90: "can only be run via a docker container, which severely limits the throughput of proteins it is able to process": No, a docker container does not in itself severely limit the throughput. What is limiting the throughput of PSORTb 3.0 is its dependence on BLAST.

- I.160-164: The sentence beginning with "When combined with..." is difficult to understand. What is the "control group" here?

- I.232-234: "a small ... but significantly lower ... reduction": did you mean "a small but significant reduction"? If not: what is the reduction lower than?

- I.248-249: "this increase in size is not accompanied by an increase in molecular weight": I don't understand this. There are more amino acids, but not a higher molecular weight? Is the increase in number of amino acids compensated by a higher proportion of small side chains?

- I.355-358: According to your own definition in the Abstract, the secretome comprises proteins secreted into the environment. In other words, periplasmic proteins should not be counted in. But then, it is not true that your estimate of the proportion of the secretome (8%) falls within the range reported in previous studies (10-30%).

- I.369-372: "There are two possible explanations..." There is a third possible explanation: that the prediction by Ayu is a false positive.
- I.403-411, the results depicted in Figure S9: I fail to see the relevance of this to subcellular location - the metric is higher in mesopelagic than in surface samples, but the trend is the same for all three locations?
- I.502-504: "As the number of...": some details about "domain annotation and synteny" would be nice.
- I.582 (and other places): You report Matthews correlation coefficient (MCC), but this measure is only defined in the two-category case, and you have three. Did you use the k-category definition described by Gorodkin (<https://doi.org/10.1016/j.compbiolchem.2004.09.006>)? In that case, cite that paper. If not, describe how you calculated one number for three categories.
- I.605-606: "Membrane-attached proteins were identified by annotation": which annotation?
- The selection of the dataset, as described in Figure S11: A sequence is only classified as "Membrane" if it has >3 predicted transmembrane domains. Doesn't that leave a number of transmembrane proteins with 1, 2 or 3 TM helices in the dataset?

Specific comments, minor:

- Abstract I.30: "The application of this powerful novel to open ocean samples": a word is missing, such as "tool" or "method"
- I.71: "shotgun metaproteomic have been used to study" it seems there is a word missing, maybe "techniques"
- I.84: "focusing only on one prediction system": should probably have been "secretion system"
- I.98: "adapter" should be "adapted"
- I.99: "Salt is" should be "Salt has"
- I.104: "osmorregulated" should be "osmoregulated"
- I.111: "isoelectric" missing "point"
- I.150: "cell wall type": please be more precise - do you mean the Gram-positive / Gram-negative distinction?
- I.159; "was distinct for periplasmic than for extracellular proteins": this does not read well, please rephrase
- I.178-179: "tolerance to temperature": high or low temperature?
- I.185: abbreviation "LRA" has not been defined
- I.186: "phylums" should be "phyla"
- I.189-190: "Figure S4 shows...": No, these are the orders shown in Figure S5. S4 shows some other orders.
- I.236-237: You mention three orders, but Figure S6 shows families. And I see no Rhodospirillales in Figure S6.
- I.238: "above mentioned" should be "mentioned above"
- I.240: "compounds" should be "elements".
- I.249: "phenomena" should be "phenomenon"
- I.256-257: "but perhaps more...": the end of this sentence is incomplete.
- I.265 and 3 other places: I would advise against using the term "sublocation" for bacteria, but instead simply say "location". In my book, "sublocation" is a sub-category of a location, such as found in eukaryotes, where "nucleus" may be subdivided into "nuclear matrix", "nucleolus", etc.
- I.446: a word missing in this sentence, probably "which".
- I.487: "UniRefKB" should be "UniProtKB".
- I.506: "excluded all other": missing "from"
- I.543-544: "Full regression results can be found in Full regression results can be found in": repetition.

- I.554: "ACC" should probably have been "AAC".
- Figure S1: the different shapes of the points are not reflected in the legend.
- Figure S3 caption: "accordin" should be "according"
- Supplementary Text 2: "described by Chou in [ref]": reference missing.

GitHub page:

- "Once all protein features have been processed, protein sublocations can be predicted with the command:" is followed by an empty line.

Reviewer #2

(Remarks to the Author)

The paper presents an analysis of the composition of extracellular proteins from marine bacteria and a new predictor to discriminate them starting from protein sequences of marine prokaryotes.

The research topic is relevant and important and the presentation is generally clear.

My main concerns regard the development and validation of the prediction method:

- Training dataset. The annotation of the dataset is only partially derived from experiments. It is unclear how many proteins are experimentally annotated and how many are assigned to a location through the predictive workflow reported in figure S1 and the following manual annotation. Rules of manual annotation should be further detailed and it should be specified how the assignment is performed when contrasting annotations are present for proteins in the same cluster (if this is the case). It is unclear whether a representative per cluster is selected and, in case, on which basis. Also, I wonder why the number of proteins from Gram+ and Gram- bacteria are quite similar despite the fact that selected Gram- organisms are largely higher.

- Splitting among training and validation sets. The splitting criteria are not clear. I can assume it has been randomly performed and in this case the complete independence between the sets is not granted, since the starting dataset contains proteins up to 50% identical (if a representative per cluster is selected). To ensure an unbiased evaluation of the method performance and generalization, proteins in the validation set should share a much lower similarity with proteins in the training set. Possibly, the results should be reported only on proteins that have been experimentally annotated, to avoid circularity between the initial annotation and the prediction (see next point).

- Input features. One of the input features (SignalP6 prediction) is also used to annotate the training/validation sets. This circularity increases the risk to unbiased evaluation of the method, if non-experimentally annotated proteins are used in the testing phase.

- Benchmark. Table 1 reports the scoring indexes of the method. It is unclear whether they are computed in cross-validation or they refer to the evaluation on the 30% of the dataset "used for further evaluation of the model". Since hyperparameters are set during the cross-validation procedure, cross-validation results risk to be overfitted. The table reports MCC values that are routinely computed for binary classification. However in this case the discriminated classes are three and it then unclear what the score refers to. Some of the considered methods (i.e., Busca) does not separate secreted proteins between periplasmic and extracellular proteins and could therefore be omitted.

Some minor point:

- the variance of the compositional data could be indicated to better evaluate the significance of their differences
- the statistics discussed in section "Differences in AAC between subcellular localizations and habitat" should be provided
- supplementary texts should be reformatted and clarified.

Version 1:

Reviewer comments:

Reviewer #1

(Remarks to the Author)

I am pleased to see that the authors have responded appropriately to almost all my comments from the first review. However, the proofreading could have been better. I still have a number of minor comments.

The process of reviewing the revised manuscript was made unnecessarily complicated by the fact that the authors had not marked the changes in the manuscript. Furthermore, there are no line numbers in the revised manuscript, making it difficult to refer to specific places.

Specific comments:

The one comment I had that is not addressed in the revision is 1.40: "Once all protein features have been processed, protein sublocations can be predicted with the command:" is followed by an empty line.'

The authors wrote that this was amended in the GitHub page, but it is not. This should arguably be the most important command in the README file, and the field is empty.

Table 1: Caption missing!

p.4: "ESKAPEE pathogens" should be explained the first time it is used.

p.4: "by a gram positive or gram negative bacteria": should be "bacterium" in singular.

p.4: "p-value 2.2e-16": equal to or less than?

p.4: "Amino acids E, G, I, K, P, T, Vare especially": space missing between "V" and "are".

p.6: "Interestingly, this increase in size is not...": I would suggest "length" instead of "size".

p.6: "but perhaps more sophisticated methods, such as HYDROPRO51 or HullRad52.": This sentence is incomplete.

p.7: BUSCA should be cited the first time it is mentioned.

p.7: "significantly over perform compared to": "overperform" should be one word.

p.8: "periplasmic (0.58) extracellular (0.46)": an "and" is missing.

p.8: "he use of SMOTE": something is missing, perhaps it should have been "Since the use of SMOTE"

p.9: SignalP6 should be cited the first time it is mentioned.

p.10: "there was a relatively high expression of these genes": It is not entirely clear what "these" refers to here.

p.11: "out of the X proteins coded by our collection": number missing!

p.12: "domains and features both sequences": an "of" is missing.

p.12: "as defined in [ref].": citation missing.

p.12 : "Figure S12 shows that": should be "Figure S13"

p.12: "the fact that their feature shapes have a non-zero feature shape": should it have been "the fact that they have a non-zero feature shape"?

p.12: "15% for testing during training, 15% for validation": I would suggest swapping the words "testing" and "validation" to conform with standard usage of these words in the field, see e.g.

https://en.wikipedia.org/wiki/Training,_validation,_and_test_data_sets

p.13: "using the R package statix89": period missing.

p.13: TMBed should be cited the first time it is mentioned.

p.13: "Generalized Applied Models (GAMs)": "Applied" should be "Additive".

p.14: "performed using graphpart 102, an homology-aware partitioner": using which threshold?

Figure S7 caption: "sulphut" should be "sulphur"

Figure S14: swap "testing" and "validation", see my last comment to p.12.

(Remarks on code availability)

Reviewer #2

(Remarks to the Author)

Authors satisfactorily replied to my concerns.

(Remarks on code availability)

RESPONSE TO REVIEWERS' COMMENTS

Reviewer #1 (Remarks to the Author):

General comment:

1.1 This manuscript describes a novel tool for predicting secreted proteins in marine bacteria. It is a very welcome addition to the prokaryotic protein sorting prediction toolbox, where the reference method, PSORTb 3.0, is more than a decade old. However, I have serious doubts about the separation between training and test data and hence about the reported predictive performance.

It is stated that 30% of the dataset were only used for evaluation of the model, and that "proteins were grouped into clusters with 50% sequence identity and at a coverage alignment of at least 80%, using the MMseqs2 software suite".

There are two problems with this. First, 50% identity and 80% coverage constitutes a rather high threshold; if this is compared to other recent subcellular location prediction methods (mainly in eukaryotes), 30% or 40% identity would be more appropriate. Second, MMseqs2 simply cannot be used in this way. Only the representative sequence of each cluster (as defined by MMseqs2) have a maximum identity of 50% to other representative sequences; this restriction does not apply to other members of the clusters. In other words, there may very well be sequences in the test set that are considerably more than 50% identical to sequences in the training set.

A proper separation between training and test can be achieved by either (i) reducing the dataset by only keeping the representative sequence of each MMseqs2 cluster, or (ii) using a homology partitioning scheme such as GraphPart (<https://doi.org/10.1093/nargab/lqad088>).

We thank the reviewer for their suggestions. We have implemented the reviewer's remarks as part of our revision to the training dataset: First, clustering is now performed with CD-HIT instead of MMSeqs, since the training dataset is now smaller, clustering identity is now 30%, following the work of other predictor methods; and only one representative sequence is chosen from each cluster. Second, we have implemented homology-aware partitioning via graphpart as part of the machine learning pipeline.

Specific comments, major:

1.2 - 1.88-90: "can only be run via a docker container, which severely limits the throughput of proteins it is able to process": No, a docker container does not in itself severely limit the throughput. What is limiting the throughput of PSORTb 3.0 is its dependence on BLAST.

Amended in the text, it It now reads: "Even pSORTb 3.0, the gold standard for subcellular localization predictions, relies on homology searches against its curated profile and protein

databases to obtain its predictions, which severely limits the throughput of proteins it is able to process.”

1.3 - I.160-164: The sentence beginning with "When combined with..." is difficult to understand. What is the "control group" here?

The control group in this case refers to the non-marine ESKAPEE proteins. We have reformatted the paragraph so it is more intelligible. The paragraph It now reads: “ When considering both combined the changes in AAC explained by subcellular location and the influence of the marine environment, we find that in 12 amino acids (L, V, F, Y, S, T, N, Q, D, E, K, R) the distribution varies following their order of exposition to the environment (Extracellular > Periplasmic > Cytoplasm), compared to 7 in the non-marine ESKAPEE group (L, Y, G, S, T, N, R).”

1.4 - I.232-234: "a small ... but significantly lower ... reduction": did you mean "a small but significant reduction"? If not: what is the reduction lower than?

We meant the former. The text It now reads: “On the one hand, there is a small (effect size 0.0161) but significant (p-value < 0.00266, kruskal-wallis test) reduction in ATP cost...”

1.5 - I.248-249: "this increase in size is not accompanied by an increase in molecular weight": I don't understand this. There are more amino acids, but not a higher molecular weight? Is the increase in number of amino acids compensated by a higher proportion of small side chains?

Exactly. The text has been modified to make this statement more clear, it It now reads: “Interestingly, this increase in size is not accompanied by an increase in molecular weight, as the amino acids more prevalent in extracellular proteins tend to contain smaller side chains.”

1.6 - I.355-358: According to your own definition in the Abstract, the secretome comprises proteins secreted into the environment. In other words, periplasmic proteins should not be counted in. But then, it is not true that your estimate of the proportion of the secretome (8%) falls within the range reported in previous studies (10-30%).

The definition of secretome varies significantly between authors and fields of study. We chose to use the definition that has been used for the study of extracellular enzymes in the marine environment, as we believe it better represents the interests of other researchers in the field and the data we used for the environmental/field application part of this study. However, the reviewer is correct in pointing out the incongruence in definitions between abstract and results. As there are no available experimental studies that determine the size of the secretome as we define it, we have decided to alter the text in the results to reflect this fact. Lines 375-376 now read: “The proportion of non-integral membrane proteins secreted

into the periplasm or to the extracellular milieu (15.5%) reported here falls in the range of the size of the secreted proteome size reported in previous studies (10-30%)”.

1.7 - I.369-372: "There are two possible explanations..." There is a third possible explanation: that the prediction by Ayu is a false positive.

It would be a possibility; however, due to the fact that many of these proteins code for domains related to extracellular activity (either because of their function or because they are a sorting signal), we are inclined to believe that they are genuine. Nonetheless, we have removed this statement from the paper, as it does not add anything relevant to the results.

1.8 - I.403-411, the results depicted in Figure S9: I fail to see the relevance of this to subcellular location - the metric is higher in mesopelagic than in surface samples, but the trend is the same for all three locations?

The relevance for this plot is that the trend is not between cellular locations but between depths (i.e., there is more production of extracellular proteins at lower depths, as it has been reported previously). We have adapted the text so that the statement and its relevance is clearer.

1.9 - I.502-504: "As the number of...": some details about "domain annotation and synteny" would be nice.

We have completely overhauled the methodology to produce the training set. In this version, classification into different versions is not based on specific domains, but on homology to proteins with manually curated subcellular location and a complete match of domain / features content and synteny to the aforementioned proteins.

1.10 - I.582 (and other places): You report Matthews correlation coefficient (MCC), but this measure is only defined in the two-category case, and you have three. Did you use the k-category definition described by Gorodkin (<https://doi.org/10.1016/j.compbiolchem.2004.09.006>)? In that case, cite that paper. If not, describe how you calculated one number for three categories.

We are indeed using the k-category definition. A citation for the paper has now been added to the manuscript.

1.11 - I.605-606: "Membrane-attached proteins were identified by annotation": which annotation?

In the previous version of the manuscript, we detected Gram-positive cell wall-attached proteins by the presence of a LPXTG domain at the C-T. However, this step was not performed as part of the analysis, since non-transmembrane proteins attached to the membrane are now predicted by Ayu as being located in their respective cellular compartment. We have removed all mentions of this step.

1.12 - The selection of the dataset, as described in Figure S11: A sequence is only classified as "Membrane" if it has >3 predicted transmembrane domains. Doesn't that leave a number of transmembrane proteins with 1, 2 or 3 TM helices in the dataset?

We are wary of including proteins that have only one Transmembrane helix, especially if it is found in the C-T (because they tend to be confused with a signal peptide). We have changed our classification methodology to include these proteins if they code for a domain that locates them to the membrane.

Specific comments, minor:

1.13 - Abstract I.30: "The application of this powerful novel to open ocean samples": a word is missing, such as "tool" or "method"

Amended in the text. It now reads: "The application of this powerful novel tool to open ocean samples..."

1.14 - I.71: "shotgun metaproteomic have been used to study" it seems there is a word missing, maybe "techniques"

Amended in the text. It now reads: "although shotgun metaproteomic approaches have been used..."

1.15 - I.84: "focusing only on one prediction system": should probably have been "secretion system"

Amended in the text accordingly.

1.16 - I.98: "adapter" should be "adapted"

Amended in the text accordingly.

1.17 - I.99: "Salt is" should be "Salt has"

Amended in the text accordingly.

1.18 - I.104: "osmorregulated" should be "osmoregulated"

Amended in the text accordingly.

1.19 - I.111: "isoelectric" missing "point"

Amended in the text accordingly.

1.20 - I.150: "cell wall type": please be more precise - do you mean the Gram-positive / Gram-negative distinction?

We indeed mean the distinction between gram-positive / gram-negative. The sentence has been edited and it now reads: "adjusting for subcellular localization and cell wall structure (that is, if the proteins were coded by a gram positive or gram negative bacteria)."

1.21 - I.159; "was distinct for periplasmic than for extracellular proteins": this does not read well, please rephrase

The sentence has been edited and it now reads: "Additionally, the marine environment affected periplasmic and extracellular proteins differently, as the AAC of extracellular proteins was influenced more strongly towards being stronger in polar, aromatic and charged amino acids."

1.22 - I.178-179: "tolerance to temperature": high or low temperature?

Both high and low temperatures. Edited in the text, it now reads: "there are multiple reports where the addition of aromatic residues to a marine protein increased its tolerance to high and low temperatures".

1.23 - I.185: abbreviation "LRA" has not been defined

Amended in the text accordingly.

1.24 - I.186: "phylums" should be "phyla"

Amended in the text accordingly.

1.25 - I.189-190: "Figure S4 shows...": No, these are the orders shown in Figure S5. S4 shows some other orders.

Figure numbers have been adapted to match the description in the text.

1.26 - I.236-237: You mention three orders, but Figure S6 shows families. And I see no Rhodospirillales in Figure S6.

The mismatch between figure and text was due to the fact the previous version of Figure S6 was submitted for revision. Figure S6 has been replaced with the intended version.

1.27 - I.238: "above mentioned" should be "mentioned above"

Amended in the text accordingly.

1.28 - I.240: "compounds" should be "elements".

Amended in the text accordingly.

1.29 - I.249: "phenomena" should be "phenomenon"

Amended in the text accordingly.

1.30 - I.256-257: "but perhaps more...": the end of this sentence is incomplete.

Amended in the text. It now reads: "but perhaps more sophisticated methods, such as HYDROPRO⁵¹ or HullRad⁵² might be able to detect differences in diffusion rates between intra and extracellular proteins."

1.31 - I.265 and 3 other places: I would advise against using the term "sublocation" for bacteria, but instead simply say "location". In my book, "sublocation" is a sub-category of a location, such as found in eukaryotes, where "nucleus" may be subdivided into "nuclear matrix", "nucleolus", etc.

Following the advice of the reviewer we have substituted all mentions of "sublocation" to "cellular location".

1.32 - I.446: a word missing in this sentence, probably "which".

Amended in the text accordingly.

1.33 - I.487: "UniRefKB" should be "UniProtKB".

Amended in the text accordingly.

1.34 - I.506: "excluded all other": missing "from"

Amended in the text accordingly.

1.35 - I.543-544: "Full regression results can be found in Full regression results can be found in": repetition.

Amended in the text accordingly.

1.36 - I.554: "ACC" should probably have been "AAC".

Amended in the text accordingly.

1.37 - Figure S1: the different shapes of the points are not reflected in the legend.

Amended in Figure S1.

1.38 - Figure S3 caption: "accordin" should be "according"

Amended in Figure S3.

1.39 - Supplementary Text 2: "described by Chou in [ref]": reference missing.

Amended in the text.

GitHub page:

1.40 - "Once all protein features have been processed, protein sublocations can be predicted with the command:" is followed by an empty line.

Amended in the GitHub page.

Reviewer #2 (Remarks to the Author):

2.1 The paper presents an analysis of the composition of extracellular proteins from marine bacteria and a new predictor to discriminate them starting from protein sequences of marine prokaryotes.

The research topic is relevant and important and the presentation is generally clear.

My main concerns regard the development and validation of the prediction method:

Training dataset. The annotation of the dataset is only partially derived from experiments. It is unclear how many proteins are experimentally annotated and how many are assigned to a location through the predictive workflow reported in figure S1 and the following manual annotation. Rules of manual annotation should be further detailed and it should be specified how the assignment is performed when contrasting annotations are present for proteins in the same cluster (if this is the case). It is unclear whether a representative per cluster is selected and, in case, on which basis. Also, I wonder why the number of proteins from Gram+ and Gram- bacteria are quite similar despite the fact that selected Gram- organisms are largely higher.

The reviewer raises an excellent point about the need for experimentally proven datasets. Although we agree that using a validation set with experimentally proven proteins would be desirable, we are limited to the study of marine prokaryotes in order to assess the effect of salt adaptation, and there are only around 40 proteins with an experimentally proven cellular location in our collection of organisms. Yet, we have now improved our manual annotation process so that it is more specific and more strict at all steps of the process (See "Data collection, annotation and curation" in Materials and Methods of the current revision). With regards to the question about the number of proteins in Gram + and Gram - bacteria, they were quite similar because we did not cluster Gram + and Gram - proteins together. This has now changed in the new version, and the new protein numbers reflect that (2346 Gram +, 15682 Gram -).

2.2 - Splitting among training and validation sets. The splitting criteria are not clear. I can assume it has been randomly performed and in this case the complete independence between the sets is not granted, since the starting dataset contains proteins up to 50% identical (if a representative per cluster is selected). To ensure an unbiased evaluation of the method performance and generalization, proteins in the

validation set should share a much lower similarity with proteins in the training set. Possibly, the results should be reported only on proteins that have been experimentally annotated, to avoid circularity between the initial annotation and the prediction (see next point).

The reviewer raises two excellent points with regards to the homology between training / validation sets and the value of a test set composed of experimentally annotated proteins. The first point was also raised by reviewer #1 and is answered in detail in question 1.1. As for the second point, although we agree that using a validation set with experimentally proven proteins would be desirable, we are limited to the study of marine prokaryotes in order to assess the effect of salt adaptation, and there are only around 40 proteins with an experimentally proven cellular location in our collection of organisms.

2.3 - Input features. One of the input features (SignalP6 prediction) is also used to annotate the training/validation sets. This circularity increases the risk to unbiased evaluation of the method, if non-experimentally annotated proteins are used in the testing phase.

We thank the reviewer for their suggestion. Although the new method for annotating the training/validation sets does not employ SignalP6 prediction, we have applied a test for circularity to confirm that this was not an issue. The outcome of this test was successful, and the results and description of the test can be found in the Materials & Methods section.

2.4 - Benchmark. Table 1 reports the scoring indexes of the method. It is unclear whether they are computed in cross-validation or they refer to the evaluation on the 30% of the dataset "used for further evaluation of the model". Since hyperparameters are set during the cross-validation procedure, cross-validation results risk to be overfitted. The table reports MCC values that are routinely computed for binary classification. However in this case the discriminated classes are three and it then unclear what the score refers to. Some of the considered methods (i.e., Busca) does not separate secreted proteins between periplasmic and extracellular proteins and could therefore be omitted.

The scoring indexes are calculated during testing with the validation dataset, only using the hyperparameter tuning and training of the models. The MCC used in this paper is Gorodkin's k-category definition, which can be used for multi classification. The text has been updated to reflect this fact, as stated in comment 1.10. Regarding the inclusion of BUSCA, although it is true that it does not distinguish between periplasmic and extracellular proteins for Gram-positive bacteria, we consider that it is important to provide as many comparisons to other state-of-the-art classifiers as possible, and it is one of the only classifiers that is readily available.

2.5 - the variance of the compositional data could be indicated to better evaluate the significance of their differences.

The statistics are provided in Supplementary Text S1. A sentence has been added to the text to better direct readers to that section of the paper.

2.6 - the statistics discussed in section "Differences in AAC between subcellular localizations and habitat" should be provided.

See question 2.5.

2.7 - supplementary texts should be reformatted and clarified.

The Supplementary texts have been reformatted for clarity.

RESPONSE TO REVIEWERS' COMMENTS

Reviewer #1 (Remarks to the Author):

I am pleased to see that the authors have responded appropriately to almost all my comments from the first review. However, the proofreading could have been better. I still have a number of minor comments.

The process of reviewing the revised manuscript was made unnecessarily complicated by the fact that the authors had not marked the changes in the manuscript. Furthermore, there are no line numbers in the revised manuscript, making it difficult to refer to specific places.

Specific comments:

The one comment I had that is not addressed in the revision is 1.40: "'Once all protein features have been processed, protein sublocations can be predicted with the command:'" is followed by an empty line.'

The authors wrote that this was amended in the GitHub page, but it is not. This should arguably be the most important command in the README file, and the field is empty.

We thank the reviewer for their kind comments and we apologize for any inconveniences caused by the manuscript (unmarked edits, no line numbers). The README file in the GitHub page has been extensively improved, going further from the changes mentioned in 1.40.

1.1: Table 1: Caption missing!

Added in table.

1.2: p.4: "ESKAPEE pathogens" should be explained the first time it is used.

Amended. Line 155 now reads: "As our control group, we chose the ESKAPEE pathogens (*Enterococcus faecium*, *Staphylococcus aureus*, *Klebsiella pneumoniae*, *Acinetobacter baumannii*, *Pseudomonas aeruginosa*, *Enterobacter spp.* and *Escherichia coli*), as they are an extensively studied group of bacteria³⁷, and none of their members are commonly found in marine environments."

1.3: p.4: "by a gram positive or gram negative bacteria": should be "bacterium" in singular.

Amended in text.

1.4: p.4: "p-value 2.2e-16": equal to or less than?

Less than 2.2e-16. Amended in text.

1.5: p.4: "Amino acids E, G, I, K, P, T, Vare especially": space missing between "V" and "are".

Amended in text.

1.6: .6: "Interestingly, this increase in size is not...": I would suggest "length" instead of "size".

Amended in text.

1.7: p.6: "but perhaps more sophisticated methods, such as HYDROPRO51 or HullRad52.": This sentence is incomplete.

Amended in text. Line 278 now reads: "We found no significant differences in molecular weight, but perhaps more sophisticated methods to calculate protein diffusion rates, such as HYDROPRO⁵² or HullRad⁵³ might uncover differences in this property between extracellular and cytoplasmic proteins."

1.8: p.7: BUSCA should be cited the first time it is mentioned.

Amended in text.

1.9: p.7: "significantly over perform compared to": "overperform" should be one word.

Amended in text.

1.10: p.8: "periplasmic (0.58) extracellular (0.46)": an "and" is missing.

Amended in text.

1.11: p.8: "he use of SMOTE": something is missing, perhaps it should have been "Since the use of SMOTE"

Amended in text. Line 357 now reads: "As the use of SMOTE only affects training time does not impact prediction running time, the SMOTE version of the multiclass implementation was kept for the final version of Ayu."

1.12: p.9: SignalP6 should be cited the first time it is mentioned.

Amended in text.

1.13: p.10: "there was a relatively high expression of these genes": It is not entirely clear what "these" refers to here.

"These" refers to extracellular proteins. Line 444 now reads "Overall, there was a relatively high expression of these genes identified as coding for secreted proteins..." in order to improve readability.

1.14: p.11: "out of the X proteins coded by our collection": number missing!

Amended in text.

1.15: p.12: "domains and features both sequences": an "of" is missing.

Amended in text.

1.16: p.12: "as defined in [ref].": citation missing.

Amended in text.

1.17: p.12 : "Figure S12 shows that": should be "Figure S13"

Amended in text.

1.18: p.12: "the fact that their feature shapes have a non-zero feature shape": should it have been "the fact that they have a non-zero feature shape"?

Indeed. Amended in text.

1.19: p.12: "15% for testing during training, 15% for validation": I would suggest swapping the words "testing" and "validation" to conform with standard usage of these words in the field, see e.g.

https://en.wikipedia.org/wiki/Training,_validation,_and_test_data_sets

Amended in text.

1.20: p.13: "using the R package statix89": period missing.

Amended in text.

1.21: p.13: TMBed should be cited the first time it is mentioned.

Amended in text.

1.22: p.13: "Generalized Applied Models (GAMs)": "Applied" should be "Additive".

Amended in text.

1.23: p.14: "performed using graphpart 102, an homology-aware partitioner": using which threshold?

Using the recommended 30% identity threshold. Line 635 now reads " The train-test partition and the splits for cross-validation were performed using graphpart¹⁰³, an homology-aware partitioner, using the recommended 30% identity threshold."

1.24: Figure S7 caption: "sulphut" should be "sulphur"

Amended in figure.

1.25: Figure S14: swap "testing" and "validation", see my last comment to p.12.

Amended in figure.

Reviewer #2 (Remarks to the Author):

Authors satisfactorily replied to my concerns.